
# Estimation of Biomass Burning Emission of NO₂ and CO from 2019-2020 Australia Fires Based on Satellite Observations

Nenghan Wan[1], Xiaozhen Xiong[2], Gerard J. Kluitenberg[1], J.M. Shawn Hutchinson[3], Robert Aiken[1], Haidong Zhao[1], Xiaomao Lin[1*]

5   [1]Department of Agronomy, Kansas Climate Center, Kansas State University, Manhattan, KS, 66502, USA
[2]NASA Langley Research Center, Hampton, VA, 23618, USA
[3]Department of Geography and Geospatial Sciences, Kansas State University, Manhattan, KS, 66502, USA

*Correspondence to*: Xiaomao Lin (xlin@ksu.edu)

25

30



**Abstract.** The bushfires that occurred in Australia in late 2019 and early 2020 were unprecedented in terms of their scale, intensity, and impacts. Using nitrogen dioxide ($NO_2$) and carbon monoxide (CO) data measured by the Tropospheric Monitoring Instrument (TROPOMI), together with fire counts and fire radiative power (FRP) from MODIS, we analyzed the temporal and spatial variation of $NO_2$ and CO column densities over three selected areas covering savanna and temperate forest vegetation. The $\Delta NO_2/\Delta CO$ emission ratio and emission factor were also estimated. The $\Delta NO_2/\Delta CO$ emission ratio was found to be $1.5 \pm 1.2$ for temperate forest fire and ranged from $2 \pm 1.3$ to $2.8 \pm 1.8$ for savanna fire. For savanna and temperate forest fires, satellited-derived $NO_x$ emission factors are $1.29$ g kg$^{-1}$ and $1.2$ g kg$^{-1}$ separately, while CO emission factors are $62.34$ and $112.5$ g kg$^{-1}$. This study demonstrates that the large-scale emission ratio from the TROPOMI satellite for different biomass burnings can help identify the relative contribution of smoldering and flaming activities in a large region and their impacts on the regional atmospheric composition and air quality.





# 1 Introduction

As a consequence of climate change, extreme climatic conditions are conducive to large wildfires around the world, resulting in extensive social, economic, and environmental impacts (Bowman et al., 2017; Filkov et al., 2020). The year 2019 was the warmest and driest year on record to date in Australia (Abram et al., 2021). The high temperature aggravated the impact of low rainfall that led to low soil moisture conditions. Recently it was reported that the strong positive Indian Ocean Dipole was one of the main influences on Australia's climate in 2019 (Annual Australian Climate Statement 2019, 2022), leading to a very low rainfall across Australia. High temperatures, combined with low rainfall and high winds further exacerbated evaporative demand, resulting in canopy dieback and increasing high fire danger indices (Boer et al., 2020; Nolan et al., 2020; Abram et al., 2021). It was Australia's record-breaking temperature and extremely low precipitation in 2019 and 2020 that caused these unprecedented fire disasters (Abram et al., 2021) which also resulted in significant ecological, social, and economic impacts. These mega-fires in 2019 and 2020 burned more than 8 million hectares of vegetation including more than 70% of forests, woodlands, and shrublands, and 816 native vascular plant species across the south-east of the continent (Godfree et al., 2021). Thirty-three lives were lost and more than 3,000 homes destroyed as a direct result of the fires (Filkov et al., 2020), while approximately 417 perished and 3,151 hospitalizations occurred as a result of smoke inhalation (Borchers et al., 2020). The direct economic loss was estimated at A\$40 billion (Wilkie, 2020).

Global fire events are considered to be the largest source of global carbon emissions, especially in grasslands and savannas (44%) and woodlands (16%) (van der Werf et al., 2010). Also, the open biomass burning produced 20% of global nitrogen oxides ($NO_X$) and one-third to one-half of carbon monoxide (CO) emissions (Wiedinmyer et al., 2011). The $NO_x$ undergoes smog photochemistry and converts to Ozone ($O_3$) leading to increased tropospheric $O_3$, whereas CO is the leading sink of the hydroxyl radical (OH) and one of the precursors to tropospheric $O_3$ (Fowler et al., 2008). Emission ratios (ER) defined as the ratio of an excess trace gas concentration ($\Delta X$, i.e., the mixing ratio of species $X$) and the excess concentration of a reference gas ($\Delta Y$) have been widely used to characterize combustion over large fire source regions (van der Werf et al., 2017, 2020). The amount of substances emitted from the burning of a particular type of land cover depends on the fuel type and completeness of combustion, for example, a relatively large amount of $NO_2$ is emitted during hotter and cleaner flaming combustion while a larger quantity of CO is emitted during the smoldering combustion phase. Therefore, the emission ratio metric can be considered as a proxy for combustion efficiency to distinguish flaming from smoldering combustion (Andreae and Merlet, 2001). Previous studies related to CO and $NO_2$ emissions were reported from anthropogenic (e.g., vehicles emission in urban regions), fossil fuel (e.g., coal and gas-fired power plant), and wildfire sectors based on surface and satellite observations (Zhao et al., 2011; Konovalov et al., 2016; Lama et al., 2019). Besides ER, emission factor (EF) is another widely used metric to provide emission information which is defined as the amount of gas released per kg of dry fuel burned (g kg$^{-1}$). It varies greatly based on individual fire conditions and fuel types. Current estimates of EFs are primarily based on laboratory studies or field measurements in limited spatial and temporal coverage (Roberts et al., 2020; Lindaas et





al., 2021). Satellite remote sensing instruments can eliminate those difficulties and obtain information on emissions from burning conditions and fuel types over large regions. The TROPOspheric Monitoring Instrument (TROPOMI) is the satellite instrument onboard the Copernicus Sentinel 5 Precursor launched by the European Space Agency and the overpass time is about 1 PM local time (Veefkind et al., 2012). The TROPOMI has demonstrated improved accuracy and high spatial resolution that facilitate investigations of trace gases from space compared to other sensors such as the Ozone Monitoring Instrument and Measurement of Pollution in the Troposphere (van der Velde et al., 2020).

Burning in Australia is responsible for 14.4% of the global annual burnt area although the land of Australia only accounts for 6% of the Earth's land area (Giglio et al., 2013). Most of these fires occurred in the semi-arid and tropical savannas that cover the northern part of the continent (Russell-smitha et al., 2007), but large bushfires also occurred in the temperate forests of southeast Australia (Cai et al., 2009). Through a multiple-year surface observation, the annual pattern of some trace gas emissions (e.g., CO) has been identified and specific emission ratios that are based on carbon monoxide (i.e., $CH_2O/CO$, $C_2H_2/CO$, $C_2H_6/CO$) from Australian savanna fires were investigated (Paton-Walsh et al., 2010; Smith et al., 2014; Desservettaz et al., 2017). However, studies related to emissions from temperate forest fires in Australia are relatively seldom (Paton-Walsh et al., 2010; Possell et al., 2015; Guérette et al., 2018) and few studies have documented $NO_2$ and CO emissions from Australian savanna and temperate forest fires over large regions.

Therefore, the objective of this study is to characterize the emission ratio and emission factor of $NO_2$ and CO over large savanna and temperate forest fires in Australia in 2019 and 2020 using TROPOMI satellite observation. Our paper structure is as follows: Sections 2 and 3 describe the datasets and methods used. In Section 4, we report the fire intensity, and daily maximum and mean $NO_2$ and CO column densities observed over 3 months in 2019 and 2020 (i.e., 1 November 2019 to 31 January 2020) over fire hotspot regions. The emission ratios of $NO_2$ relative to CO for savanna and temperate forest fires are also examined. Finally, we estimated the EF using satellite-derived $NO_x$ and CO emissions. Section 5 is a summary and conclusion.

## 2 Data Used

### 2.1 GFED4s database

The Global Fire Emission Database version 4 with small fires (GFED4s) provides global estimates of monthly and daily burned area, emissions, and fractional contributions of different fire types with $0.25º \times 0.25º$ spatial resolution (Randerson et al., 2012). This database uses the Moderate Resolution Imaging Spectroradiometer (MODIS) Collection 5.1 MCD64A1 burned area product and includes small fires for emission estimates (Giglio et al., 2013). Six fuel classifications are estimated using land cover type product from MODIS and the University of Maryland classification scheme in the GFED4s



database, including temperate forest, boreal forest, deforested and degraded land, peatland, agricultural waste burning, and herbaceous fuel type which is composed of shrubland, savanna and grassland (van der Werf et al., 2017). In our study, the vegetation fires that happened in Australia from November 2019 to January 2020 were classified as the savanna and temperate forest fires based on GFED4s. Highlighted in Figure 1 are the three areas of interest employed in this study. There

were selected from stronger biomass burning from November 2019 to January 2020 according to Godfree et al. (2021). The three selected areas include two savanna fire areas in northwestern (Area 1) and northeastern (Area 2) Australia, as well as an area with both savanna and temperate forest fires in southeast (Area 3) Australia (Fig. 1).

## 2.2 TROPOMI CO, NO₂, and aerosol layer height (ALH) data

The total column density of CO from TROPOMI was estimated from spectral radiance measurements from the shortwave to

infrared spectral ranges around 2.3 μm that are sensitive to CO absorption with a daily $5.5 \times 7$ km$^2$ resolution (Landgraf et al., 2016; Borsdorff et al., 2018). Previous studies have shown that TROPOMI was able to capture the variability of daily CO as a result of atmospheric transport of pollution (Borsdorff et al., 2018; Schneising et al., 2020). The NO₂ tropospheric column density is detected from TROPOMI's 405 – 465 nm wavelength bands with a $5.5 \times 3.5$ km$^2$ resolution. Although there exists a negative bias of approximately 30% in the lower tropospheric columns because of cloud pressure and the *priori*

NO₂ profile used in air mass factor calculations (Lambert et al., 2018), it is still appropriate to use TROPOMI NO₂ to quantify fire burning efficiency (Lama et al., 2019; van der Velde et al., 2020). Different algorithms are used to estimate NO₂ and CO in TROPOMI instrument channels which also provide quality assurance values (i.e., qa_value) to help filter raw data under unclear sky conditions and/or other problematic retrievals. In our study, we collected CO retrievals with a qa_value larger than 0.5 and NO₂ retrievals with a qa_value larger than 0.75. The CO total column density and NO₂ tropospheric

column density were then converted to units of moles per square meter (mol m$^{-2}$) and millimoles per square meter (mmol m$^{-2}$), respectively. The TROPOMI also provides aerosol layer height (ALH) data that are based on the O₂ absorption band at near-infrared wavelengths (Graaf et al., 2019). ALH data were used to define the main vertical wind layer which was required for the emission estimation procedure described in Section 3.2. We collected ALH data with a qa_value > 0.5 and re-sampled it to the same spatial resolutions as the CO and NO₂ data. All TROPOMI datasets (CO, NO₂, and ALH data)

from November 2019 through January 2020 were included because these three months were reported as the largest fires during the 2019/20 black summer fires in Southeast Australia (Abram et al., 2021). All data were then re-sampled to 0.05° × 0.05° spatial resolution through an areal weighted interpolation using the Harp package from python (Niemeijer, 2017).

## 2.3 MODIS fire radiative power (FRP) and fire events

The FRP represents the instantaneous radiative energy that is released from actively burning fires and is related to the rate of

biomass combustion (Wooster et al., 2003), the emission rate of trace gases, and aerosol emissions (Kaiser et al., 2012). The MODIS instrument is onboard both the Earth Observation System Terra and Aqua satellites of the National Aeronautics and





Space Administration and measures radiance in spectral channels to detect fires at a 1 km spatial resolution (Kaufman et al., 1998). The MODIS near real-time active fire products data (MCD14DL) were used to identify fire events from November 2019 through January 2020. For each day, fire pixels (i.e., $1 \times 1$ km$^2$ grid cells) located within a 20 km distance of one

another were aggregated into a "fire event" and a rectangular polygon region with $\pm$50 km crosswind distance and 100 km downwind distance which is large enough to include fire pixels in the group was defined to calculate emission in Section 4.3. The fire event's center was set as the average latitude and longitude of all fire pixels, weighted by each pixel's FRP which is related to trace gas emission and widely used to estimate fire intensity (Wooster et al., 2003; Li et al., 2018). We retained only fire events in which the total FRP was larger than 200 megawatts (MJ s$^{-1}$). It should be noted that MODIS does not

provide all fire event data due to cloudy days.

## 2.4 Wind

Wind fields, which include wind speed and direction, were obtained from the hourly ERA-5 reanalysis dataset from the European Center for Medium-range Weather Forecast (ECMWF). This dataset provides meteorological variables for 37 vertical layers from 1000 hPa to 1 hPa from 1979 to the present at 0.25º $\times$ 0.25º horizontal resolution (Hersbac et al., 2020).

We first selected ERA-5 wind data at TROPOMI overpass time (1 PM at local time) and interpolated wind fields data to produce 0.05º $\times$ 0.05º resolution grids. Then, the data was vertically interpolated to the averaged ALH level within each fire event. For fire events without valid ALH data, we used 850 hPa, as the average level for all selected fire events is 850 hPa.

## 3. Methods used for Calculating Emission Ratio and Emission Factor

### 3.1 Emission ratio (ER)

Excess trace species concentration ($\Delta X$) is defined as the difference between concentrations of species $X$ in the fire plume ($X_{fire}$) and in the ambient background ($X_{bg}$). Usually, $\Delta X$ is divided by a reference species ($\Delta Y$), such as CO or CO$_2$, to get the emission ratio (ER) between those two emitted compounds (i.e., $\Delta X/\Delta Y$). In our study, a similar local sampling method by van der Velde et al. (2020) was used to calculate the ER. To calculate excess gas concentration over the three selected 10º $\times$ 10º areas (Fig. 1), daily TROPOMI data were first re-sampled into a 0.05º $\times$ 0.05º spatial resolution grid. Next, co-located

NO$_2$ and CO column densities from TROPMI were obtained from locations where NO$_x$ and CO values were available from the GFED4s database in three selected areas (Fig. 1). The $X_{fire}$ plume value was calculated as the average of all selected column densities. The corresponding  ambient background $X_{bg}$ value was calculated as the average of all values inside a 5º $\times$ 5º subregion upwind of the biomass burning region but within the three 10º $\times$ 10º study areas. The background subregions were determined by visual inspection through examing the predominant direction of the individual plume. Excess NO$_2$ and

CO concentration were determined from the expressions $\Delta NO_2 = NO_{2_{fire}} - NO_{2_{bg}}$ and $\Delta CO = CO_{fire} - CO_{bg}$, respectively,





and the emission ratio was thus calculated as $ER = \Delta NO_2/\Delta CO$. Days with inadequate data coverage (when the missing area exceeded 25% of the selected area in a single day) in either the background or study areas were removed during computation.

### 3.2 Emissions from satellite measurement and emission factor (EF)

In our study, we used an integrated mass enhancement method that has been used in previous studies (Mebust et al., 2011; Adams et al., 2019; Griffin et al., 2021) to estimate downwind flux. The periods from December 2018 to January 2019 were used as the background data for both CO and $NO_2$ column densities to represent emissions under fewer fire situations. To improve background robustness for daily gas column density, raw column density values that were above the 99[th] percentile were removed and then refilled back by using the nearest neighbouring interpolations. The daily column density was then

calculated by subtracting corresponding monthly background values from raw daily column density values. Fire pixels were grouped based on distance as described in Section 2.3 and surrounding rectangles were defined. The total mass, m $(g)$, emitted by fires is the product of daily column density and area (Eq. 1).

$$m = VCD \cdot A, \tag{1}$$

VCD $(mol\ m^{-2})$ is the daily vertical column density after subtracting background values, A is the rectangle area $(m^2)$. A

line density derived from a plume traveling gaussian model over downwind under assumptions of constant wind without diffusion and deposition (Adams et al., 2019) is expressed as Eq. 2.

$$L(x) = L_0 \cdot e^{-kt} = L_0 \cdot e^{-\frac{x}{\tau\mu}}, \tag{2}$$

Where $L_0$ $(mol\ m^{-1})$ is the concentration over the fire center calculated by integrating VCD $(mol\ m^{-2})$ from $\pm 50$ km crosswind direction, the lifetime $\tau$ is the inverse of reaction rate coefficient k $(\tau = 1/k)$, $t$ is the time for emitted gas

transport from fire center to downwind distance $x$. $\mu$ is averaged wind speed at the mean ALH level in the rectangle to yield a single wind direction for the fires. $L(x)$ $(mol\ m^{-1})$ is the line density at $x$ downwind distance. The total mass $m$ also equals the integral of gas density from the fire center to $x$ distance (Eq. 3).

$$m = \int_0^x L_0 \cdot e^{-\frac{x}{\tau\mu}}\,dx = L_0 \cdot \tau \cdot \mu \cdot \left(1 - e^{-\frac{x}{\tau\mu}}\right) = L_0 \cdot \tau \cdot \frac{x}{t} \cdot \left(1 - e^{-\frac{x}{\tau\mu}}\right), \tag{3}$$

Therefore, $t = \frac{x}{\mu}$ is the residence time inside the areas from the fire center to downwind distance. $L_0 x t^{-1}$ equals to the

emission rate E $(g\ s^{-1})$. Therefore, the relationship between total mass and the emission rate can be expressed as:

$$E = \frac{m}{\tau \cdot \left(1 - e^{-\frac{x}{\tau\mu}}\right)}, \tag{4}$$



In this study, the downwind distance $x$ is set as 20 km ($x_c$) based on previous studies (Adams et al., 2019; Griffin et al., 2021), therefore the area in Eq. 1 is the area of 20 km downwind distance. At last, we used Eq. 4 to estimate the emission rate with constant wind and estimated lifetime by using Eq. 2. Figure 2 (a) - (c) shows an example of calculating emission with a fire event that occurred in area 3 of southeastern Australia (29.2 °S, 151.5 °E) on 6 November 2019. We derived CO and $NO_2$ emission flux in $g\ s^{-1}$ based on Eq. 4 and a ratio of $NO_2/NO_x$ of 0.75 was used to convert $NO_2$ to $NO_x$. Previous studies (Yurganov et al., 2011; R'Honi et al., 2013; Whitburn et al., 2015) indicated a 7-day or 14-day effective lifetime for CO, so a 7-day effective lifetime was used in our study determined through a sensitivity test discussed in section 4.3. For the short lifetime $NO_2$, Mebust et al. (2011) assumed a 2-hour effective lifetime based on the fitted lifetimes from the OMI tropospheric $NO_2$ columns while Tanimoto et al. (2015) used 2 hours or 6 hours as the effective lifetimes. In our study, Eq. 2 was used to estimate the $NO_2$ lifetime by fitting an exponential to L(x) as a function of downwind distance and wind speed. Finally, we used the emission coefficient (g MJ$^{-1}$), an energy-based coefficient, which is defined as the mass of pollutants emitted per unit of radiative energy. The emission coefficient was estimated as a slope of a linear relationship with an intercept fixed at zero between emission estimates and FRP (Vermote et al., 2009). For temperate and savanna fires, we converted regression emission coefficients to the EFs using an energy-to-mass factor of 0.41 ± 0.04 kg MJ$^{-1}$, which is the average of the 0.368 ± 0.015 kg MJ$^{-1}$ and 0.453 ± 0.068 kg MJ$^{-1}$ values found in studies (Wooster et al., 2005; Freeborn et al., 2008; Vermote et al., 2009).

## 4. Results and Discussion

### 4.1 Temporal evolution of fire intensity and total column density

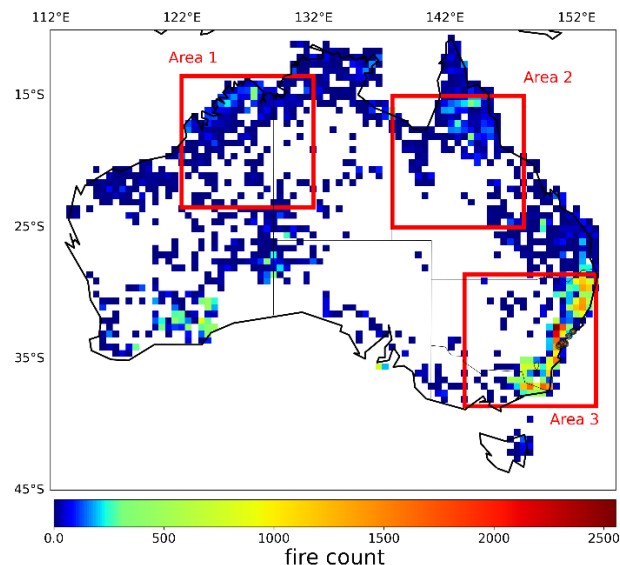





**Figure 1: Total fire counts from November 2019 to January 2020 at 0.25º × 0.25º resolution. Three 10º × 10º (latitude × longitude) areas indicated regions of interest in this study.**

The majority of fire-affected regions during these extreme fire events were located in area 3 in southeast Australia (Fig. 1) where cumulative fire counts exceeded 1,000. Fire frequencies were much lower in areas 1 and 2 where cumulative fire counts rarely approached 250. The fire-affected areas were dominantly located either in far northern oceanic boundaries of areas 1 and 2 or in the south-eastern oceanic boundary of area 3 (Fig. 1). From fire data product of MCD14DL, the daily FRP observations showed a few distinct peaks of fire events, including three weeks from November 3$^{rd}$ to 25$^{th}$ in area 1 and a second three-week period for area 2 from December 7$^{th}$ to 26$^{th}$. For area 3, there were two short FRP peaks in November and early January. The highest FRPs during these three months were $4.18 \times 10^4$, $3.27 \times 10^4$, $9.93 \times 10^5$ MJ s$^{-1}$ for area 1, area 2, and area 3, respectively. The most intensive fire events in area 1 were observed in November 2019, in area 2 in December 2019, and in area 3 in January 2020 (Fig. 3). Within these three months, both NO$_2$ and CO column density distributions showed a larger mean value for each month over area 3 compared to the other two study regions (Fig. 4). These higher NO$_2$ and CO column density observations reflect the larger FRP over area 3 (Figs. 3 and 4). As expected, the daily maximum NO$_2$ column density in area 3 was nearly double that of the other two areas (Fig. 5a) but their mean values were comparable (Fig. 5c), indicating highly fluctuated NO$_2$ densities on a fire day. On the other hand, daily maximum CO column density was nearly 10 times higher in area 3 than those estimated for area 1 and area 2 (Fig. 5b), suggesting the role of different fuel and fire combustion types. The maximum daily column densities were observed as 1.1 mmol m$^{-2}$ for NO$_2$ and 2.3 mol m$^{-2}$ for CO on 4 January in area 3. For the daily mean total column densities, both NO$_2$ and CO are significantly different for all three areas under the two-sample t-test. Again, the daily mean CO was more sensitive to the FRP compared to NO$_2$ (Fig. 5d). In addition, significant increases in CO and NO$_2$ mean values in area 3 were observed in early January, which certainly was associated with the large FRP values that occurred on 30 December 2019, and 4 January 2020 (Fig. 3) by MODIS satellites.

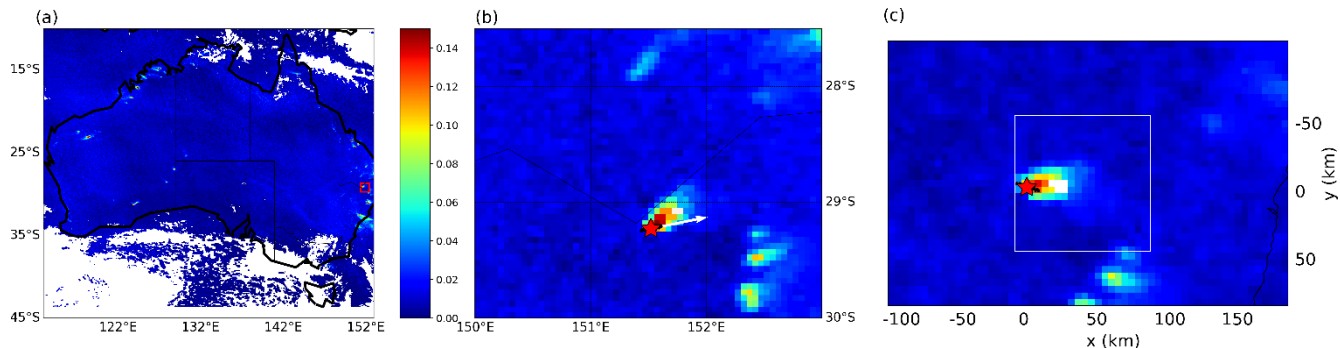

**Figure 2: An example of emission analysis for a fire event, with MODIS fire pixels indicated (black points) and the center of the fire event indicated by a red star. (a) Map of TROPOMI NO$_2$ column density over Australia on 6**





November 2019. The red box in southeast Australia is the fire event location. (b) The original TROPOMI NO₂

column density with the wind direction indicated by a white arrow. (c) The excess NO₂ after 1) removing background

260    column density from original NO₂ and 2) rotating the entire pixels examined to align with the wind direction, thus a

20 km downwind distance area selected was used to estimate the NO₂ emission.

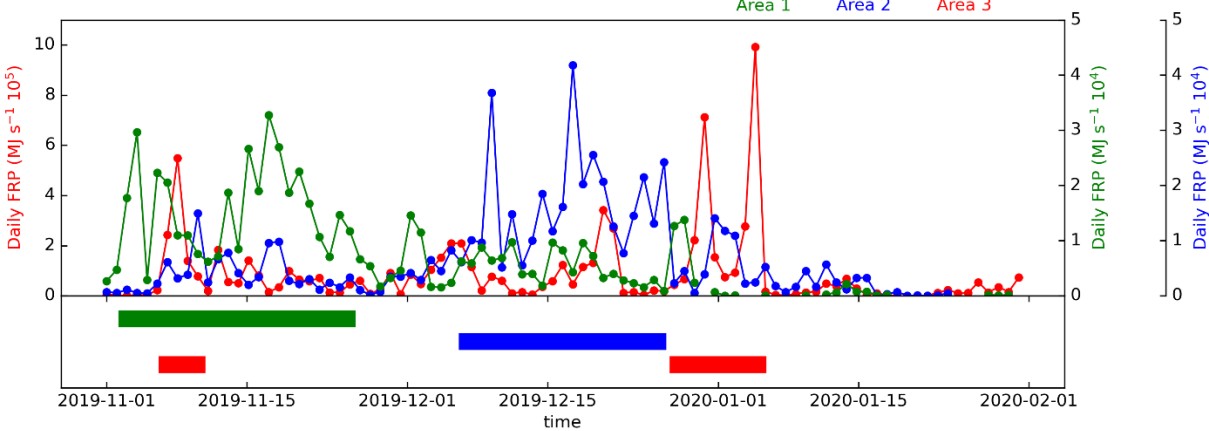

**Figure 3:** Daily fire radiative power (FRP) from November 2019 to January 2020 for area 1 (green), area 2 (blue),

265    **and area 3 (red). Several distinct periods are highlighted to show the significant increase in FRP, covering 3–25**

November (area 1), 7–26 December (area 2), 7–10 November (area 3), and 29 December–5 January (area 3).

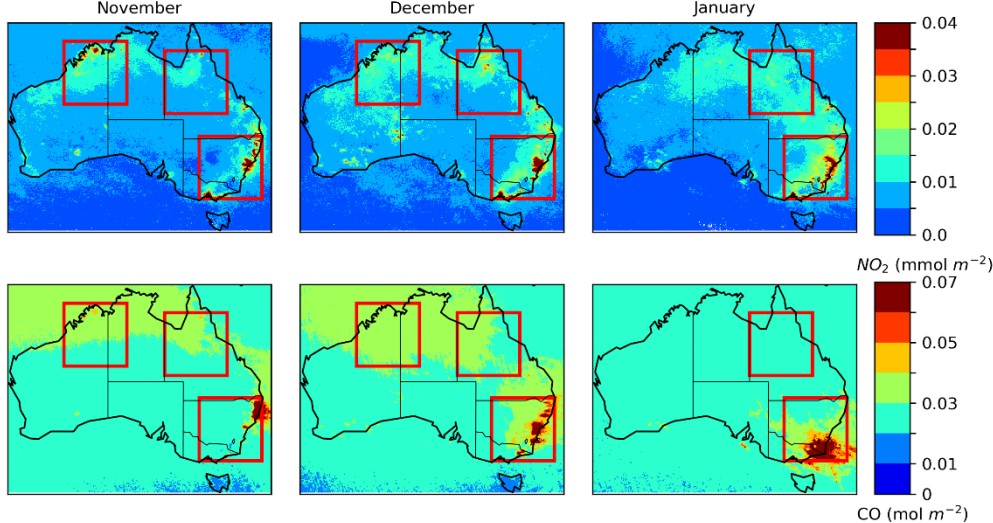

**Figure 4:** Monthly average NO₂ (upper panel) and CO (lower panel) column density from November 2019 to January

270    2020. Three 10° × 10° (latitude × longitude) areas indicated regions of interest in this study.

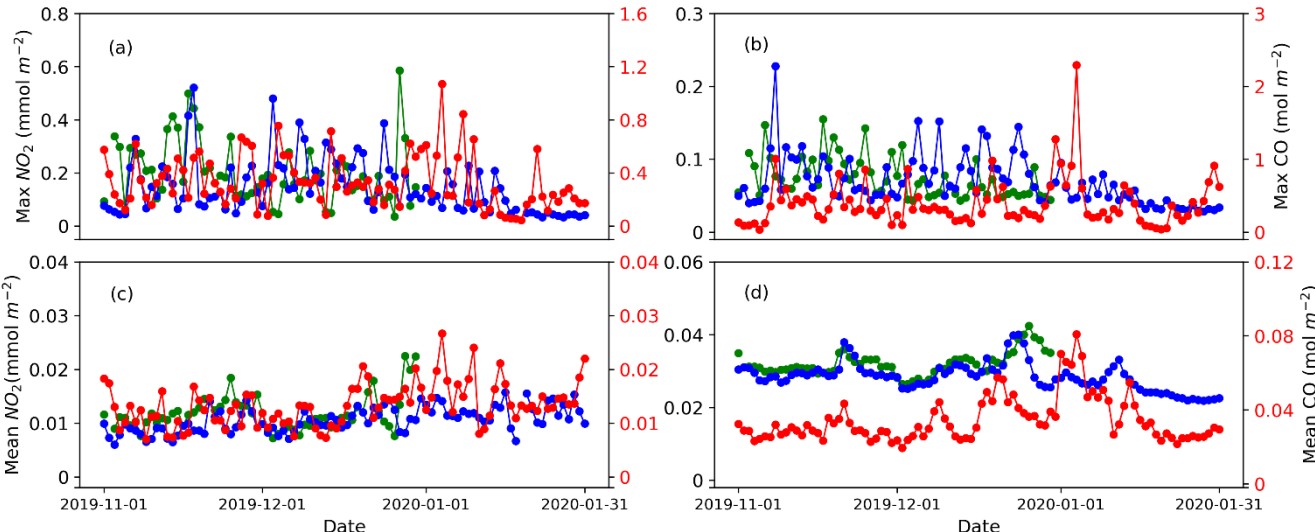

**Figure 5: Time series of daily maximum NO₂ (a) and CO (b) total column densities from November 2019 to January 2020 as well as daily mean NO₂ (c) and CO (d) for three highlighted areas: area 1 (green), area 2 (blue), and area 3 (red). Both areas 1 and 2 are displayed by the left Y axis and the area 3 are displayed by red colours of the right Y axis.**

## 4.2 Emission ratio (ER) in savanna and temperate forest

Unlike directly calculating gas concentrations, the excess gas concentration (expressed as $\Delta X$) removes the impact of potentially varying amounts of background concentration and thus represents the gas emissions related to fire activities. The averaged ERs derived from savanna fires were 2.3, 2.8, and 2.0 for areas 1, 2, and 3, respectively. The ER for temperate forests in area 3 was, on average, 1.5 during the three months of this study period (Fig. 6). As expected, $\Delta NO_2$ and $\Delta CO$ both increased with increasing FRP (high FRP periods were highlighted in Fig. 3 to correspond to points with black edge markers shown in Fig. 6) for both savanna and temperate forest-dominated landscapes, but there was a clear distinction between savanna and temperate forest fires. For the savanna fires, $\Delta NO_2$ could approach 0.05 mmol m⁻² whereas changes in $\Delta CO$ were much less at 0.03 mol m⁻² across all three study areas. However, $\Delta NO_2$ (up to 0.08 mol m⁻²) and $\Delta CO$ (up to 0.08 mol m⁻²) for temperate forest fires in area 3 were both larger in magnitude and variability. $\Delta CO$ and $\Delta NO_2$ emissions in temperate forest regions showed a larger enhancement compared to savanna fires. The $\Delta NO_2$ and $\Delta CO$ in temperate forests exceeded those in savanna fires within the same region because temperate forest fuels consisted mainly of eucalyptus trees (Godfree et al., 2021). The relatively high $\Delta NO_2$ and small $\Delta CO$ in the savanna portion of the three burning areas showed that the flaming combustion phase was dominant in savanna fires as this phase tends to produce higher NO₂ as previous research



showed (Andreae and Merlet, 2001). The day-to-day variability in $\Delta NO_2$ was larger than $\Delta CO$. The $\Delta CO$ emission ranged from 0 to 0.08 mol m$^{-2}$ whereas $\Delta NO_2$ emission changed from 0 to 0.08 mmol m$^{-2}$. Compared to van der Velde et al. (2020) who estimated $\Delta NO_2/\Delta CO$ ERs ranged between 3.58 and 6.2 for savanna fires, the ER values in our study were lower and ranged between 2 and 2.8. The ER in temperate forest combustion reported here (1.5) was also lower than the results from Young et al. (2011), which was $5 \pm 2$ mmol mol$^{-1}$, suggesting a complex interaction between dominant vegetation and local atmospheric turbulence during fire events. Although there are uncertainties from TROPOMI, there were distinct ERs clear resulting from savanna and temperate forest combustion (Fig. 7). This result suggests that temperate forest fires emitted larger CO per unit NO$_2$ compared to savanna fires, indicating less efficient combustion in temperate forest fires than in savanna fires (Fig. 7).

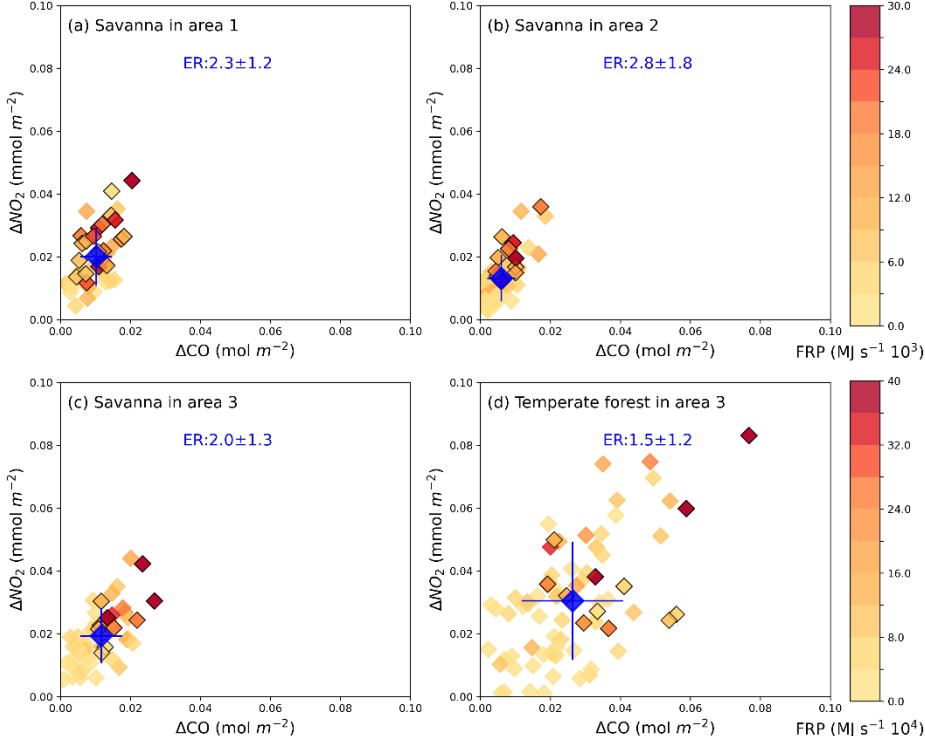

**Figure 6: The relationship between daily ΔCO (mol m$^{-2}$) and daily ΔNO₂ (mmol m$^{-2}$) in Savanna regions (a for area 1, b for area 2, and c for area 3) and temperate forest regions (d for area 3 only). The colour bars are coded by daily FRP, data points with black edges are the days with high FRP (highlighted periods) in Fig. 4. The blue markers represent the monthly average relationship between ΔCO and ΔNO₂ with day-to-day variabilities shown represented by the error bars. ER stands for the grand emission ratio expressed by grand mean plus and minus one standard deviation.**




One possible reason for different ER values was the different land surface sensitivities of TROPOMI in CO and $NO_2$ measurements (Val Martin et al., 2018; van der Velde et al., 2020). Previous studies have shown that tropospheric $NO_2$ measurement was less sensitive to sources in the planetary boundary layer than CO measurements, which causes the underestimated $\Delta NO_2$ (Borsdorff et al., 2018; van der Velde et al., 2020). A second source is the highly reactive property of $NO_2$. The short lifetime of $NO_2$ makes the daily values underestimated compared to the CO measurement which gas has a

relatively long lifetime. In addition, the natural variability of atmospheric composition (e.g., tropospheric $O_3$, water vapor) and different measurement techniques may contribute to the measurement uncertainty.

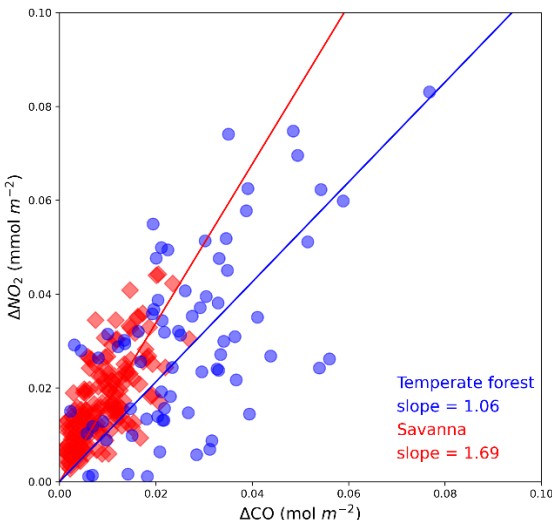

**Figure 7: The relationship between daily $\Delta CO$ (mol m$^{-2}$) and daily $\Delta NO_2$ (mmol m$^{-2}$) was derived from TROPOMI for all regions. The slope of linear fit with an intercept at zero represents the combustion efficiency of different fire types.**


### 4.3 Satellite-derived emission factor (EF)

After deriving the $NO_2$ and CO emissions for fire events, we calculated the emission coefficient (g MJ$^{-1}$) using satellite-derived emissions and FRP. The 95 % confidence intervals of the slope were computed based on the student's t-distribution test. Figure 8 shows the relationship between TROPOMI-derived $NO_x$, CO emissions and MODIS FRP for savanna and

temperate forest fires in three areas. FRP explains 42% to 60% variance in $NO_x$ emissions with the highest $R^2$ in temperate fires in area 3 and lowest in savanna fires in area 1. For CO emission, FRP explained 42% to 47% variance with the highest $R^2$ in savanna fires and the lowest in temperate fires. The variability may relate to multiple uncertainties including the satellite retrieval and emission estimate approach as we discussed below. Comparing different fire types, the $NO_x$ emission coefficient in savanna fires in area 2 is the largest (0.53 g MJ$^{-1}$), with 95% confidence intervals of 0.44 – 0.61 g MJ$^{-1}$, CO

emission coefficient in temperate forest fires in area 3 is the largest (57.67 g MJ$^{-1}$), with 95% confidence intervals of 57.06 – 63.28 g MJ$^{-1}$.



To compare with previous studies, we converted emission coefficients to EFs by applying a conversion factor K= 0.41 kg MJ$^{-1}$ (Vermote et al., 2009). For NO$_x$, the satellite-derived EFs range from 1 to 1.29 g kg$^{-1}$ in savanna fires which are

agreeable with previous studies (1.36 g kg$^{-1}$) in the Jin et al. (2021) using original TROPOMI NO$_2$ data without updating a priori profile but much lower than the work in Andreae (2.5±1.3 g kg$^{-1}$) (2019) that presented the updated compilation of EFs over the past 20 years. For temperate forests, the satellite-derived EF$_{NOx}$ is 1.2 g kg$^{-1}$, which is also less than Andreae's EFs (3 g kg$^{-1}$) (2019). For CO, the satellite-derived EF$_{CO}$ in savanna fires ranges from 57.1 to 62.34 g kg$^{-1}$ and is lower than Andreae's EFs (69 ± 20 g kg$^{-1}$) (2019) but in the range of the field measurement (ranging 15 to 147 g/kg) from SAFIRED

campaign savanna fires in Australia (Desservettaz et al., 2017) Our satellite-derived EF$_{CO}$ in temperate forest fires is 112.5 g kg$^{-1}$ which is close to Andreae's EFs (113 ± 50 g kg$^{-1}$) (2019) and Guérette's filed measurement (ranging 101 to 118 g kg$^{-1}$) in Australia temperate forest fires (Guérette et al., 2018).

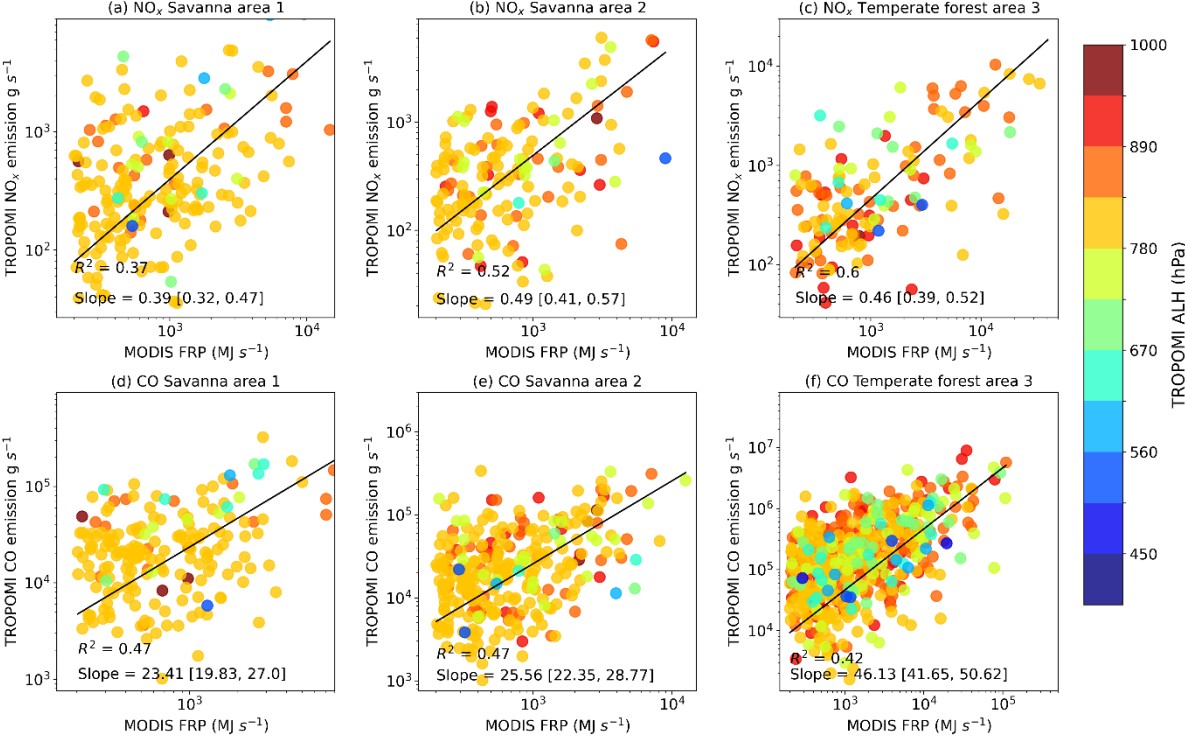

**Figure 8: Scatter plots of TROPOMI-derived NO$_x$ and CO emissions (g s$^{-1}$) versus MODIS FRP in Savanna regions (a, d for area 1, b, e for area 2) and temperate forest regions (c, f for area 3). The black line indicates the regression line estimated from ordinary least squares with the intercept fixed at zero. Slopes are shown with a 95% Confidence Interval. The color represents the TROPOMI ALH of the corresponding fire events. Emissions and FRP are on log scales.**






Our NOₓ and CO EFs are smaller than previous studies, especially EF$_{NOx}$. One source of this variance is because of aerosol smoke on the CO and NO$_2$ volume column densities. Hirsch et al. (2021) found that unprecedented bushfires in Australia caused record-breaking levels of aerosols, as TROPOMI CO values were monitored using radiances in the shortwave infrared bands so that the smoke aerosol does not have a strong effect on measurements. Schneising et al. (2020) show that

the uncertainty due to smoke aerosol during several Californian wildfires was about 5%. However, smoke aerosols have always affected TROPOMI NO$_2$ observations in the ultraviolet-visible region when estimating fire emissions. Previous studies showed an implicit aerosol correction can be applied to retrieval algorithms (Griffin et al., 2021) and without this correction, a bias of more than 40% over polluted regions could be introduced (Lorente et al., 2017), suggesting that the estimated daily CO net emission was much more accurate than NO$_2$. The uncertainty in the satellite emission method can

also cause the variance, one is the lifetime used in emission estimation. Figure 9 shows the example of fits for NO$_2$ in area 2, and the embedded histogram shows the frequency distribution of NO$_2$ lifetime ranging from 1 to 4 hours over all three areas. Thus, an average of 2.5 hours for NO$_2$ selected in our computation was optimal for calculating emission. To test the uncertainty related to different lifetime choices, the Adams et al.'s (2019) test was followed. Fluxes were recalculated by replacing the default lifetime ($\tau_{NO_2}$ = 2.5 hours, $\tau_{CO}$=7 days) into alternate lifetimes ($\tau_{NO_{2\,lower}}$ = 1 hour, $\tau_{NO_{2\,upper}}$ = 4 hours,

and $\tau_{CO_{lower}}$= 14 days), then the percent difference between EFs were calculated. The largest deviation from the default settings was defined as the uncertainty (Adams et al., 2019). For CO, the uncertainty was smaller based on the 14 days lifetime (less than 1%) while the uncertainty was larger for NO$_2$ (43% in savanna fires in area 1) based on the largest 4-hour lifetime.

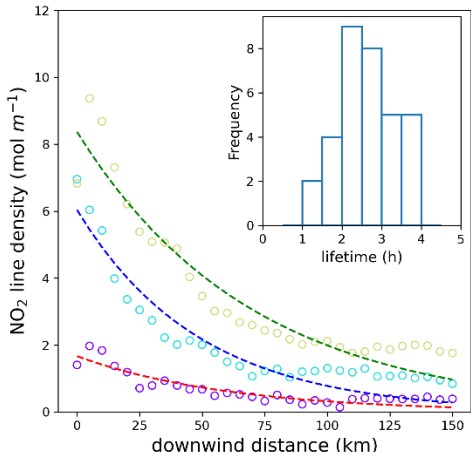

**Figure 9: NO$_2$ line density decay curves along with 150 km downwind distance in area 2. The embedded histogram shows the frequency distribution of NO$_2$ lifetime estimated from all three areas.**





## 5 Summary and Conclusions

The 2019-2020 black summer fires in Australia emitted large amounts of trace gases and aerosols. In this study, we focused
on the analysis of two trace gases: CO and $NO_2$. Based on the total columns (mean and maximum) from TROPOMI
observations and the fire intensity from MODIS in late 2019 to early 2020, we estimated the ERs of $NO_2$ relative to CO for
each day over three selected areas with savanna and temperate forest vegetation. For temperate forest fires, the ER was $1.5 \pm$
1.2 which is consistent with previous studies. For savanna vegetation fires, the ER ranged from $2 \pm 1.3$ to $2.8 \pm 1.8$, which
was slightly lower compared to other studies. These differences could be traced back to different measurement techniques
used, their spatial resolutions, nonlinear sensitivities to gas densities in the boundary layer, and larger $NO_2$ natural variability
due to its short lifetime, all of which suggest that further validation of satellite products and investigations of more cases are
required.

Using the methods from Mebust et al. (2011) and Adams et al. (2019), net emission fluxes were estimated by using a 14-day
CO effective lifetime and a 2.5-hour $NO_2$ effective lifetime, and EFs were calculated. The TROPOMI-derived $NO_x$ EFs were
1.29 g $kg^{-1}$ and 1.2 g $kg^{-1}$ for savanna and temperate forest fires which are lower than previous studies while the CO EFs
were 62.34 g $kg^{-1}$ for savanna fires and 112.5 g $kg^{-1}$ for the temperate forest. Our study on both savanna and temperate forest
fire emissions demonstrates the capability and limitations of TROPOMI data for the study of the regional variability of
combustion characteristics and their impacts on regional atmospheric composition and air quality.






## Data availability

TROPOMI CO, NO$_2$ and ALH data are available from NASA Goddard Earth Sciences (GES) Data and Information Services Center (DISC, https://disc.gsfc.nasa.gov/datasets/). MODIS FRP data are available from NASA Earth Data Fire Information for Resource Management Systems (https://earthdata.nasa.gov/earth-observation-data/near-real-time/firms). GFED4s fire emissions are available from https://www.geo.vu.nl/~gwerf/GFED/GFED4/. Wind data from the European Center for Medium-range Weather Forecast (ECMWF) is available at https://cds.climate.copernicus.eu/cdsapp#!/dataset/reanalysis-era5-pressure-levels-preliminary-back-extension

## Author contribution

NW worked on the emission estimate methodology. HZ helped to interpret the satellite datasets. XX and LX conceived the structure of the paper. NW prepared the paper, and all authors contributed to the discussion and revision of the paper.

## Competing interests

The authors declare that they have no conflict of interest.

## Acknowledgments

This study was supported in part by the U.S. Department of Agriculture, National Institute of Food and Agriculture (grant no. 2016-68007-25066). The contribution number of this manuscript is 22-275-J. We thank Dallas Staley for her outstanding contribution in editing and finalizing the paper. Her work continues to be at the highest professional level.



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
