# Peer review of "Estimation of Biomass Burning Emission of NO2 and CO from 2019-2020 Australia Fires Based on Satellite Observations"

_Atmospheric Chemistry and Physics, 2022_

## Author Comment (AC1)

**Point-by-point Responses to Reviewer #1 for # MS No.: acp-2022-447**

The authors have used regular fonts for the Referee's comments (which might be divided into two or multiple comments), blue fonts for our responses, and red fonts with quotation marks to show the revised text. In this point-by-point response, figures for the response are numbered, as follows. The 'R' stands for 'Response'. For example, "Fig. R1" means it is the first figure in response to the Reviewer.

**Reviewer 1 # Comments**

The study presents an analysis of satellite-measured CO and $NO_2$ for the Australian 2019/2020 wildfire season using TROPOMI. The authors use the satellite-measured enhancement ratio near fires in southeast Australia, as well as for two northern Australian regions, to derive a proxy for combustion efficiency and calculate emission factors for large regions.

Overall the manuscript is well written and the analysis rigorous. Satellites provide an opportunity to determine emissions over larger areas than field campaigns or laboratory studies. The results and methodologies presented here have the potential to be applied to other regions and for other years. I have several comments to be addressed below, roughly in order of importance.

**Main Comments:**

1. The temporal period (November 2019- January 2020) chosen was aligned with the maximum burning in southeast Australia (Area 3). The fire season in Areas 1 and 2 are usually different to Area 3. The peak burning season in Area 1 and Area 2 occur in September – October, while Area 1 usually peaks December – January (Russell-Smith et al., 2007, https://doi.org/10.1071/WF07018). The peak wildfire emissions in Areas 1 and 2 were likely not captured in this study. Consequently, the comparison of Area 3 and Areas 1 and 2 (e.g. on page 9) compares a mid-burning season (SE) with a late-burning season (NW). Additionally, 2019 was an an usually low year for biomass burning in Northern Australia, so the season may not be representative of the region on average. Please clarify the motivation for including Areas 1 and 2 and the timing in this study.

**Response:** Thank you for your comments. The objective of our study was to characterize the emission ratio and emission factor of $NO_2$ and CO over savanna and temperate forest fires in Australia. We included Areas 1 and 2 because they are the main areas where savanna fires occurred (Russell-Smith and Yates, 2007), whilst the fires that occur in Area 3 were mainly temperate forest fires.

The reviewer makes an excellent point about the fire season in Areas 1 and 2 being different than in Area 3. Our original study period excluded some important fire events in Area 1 during October 2019. That was an oversight on our part. To address this issue, we have adjusted the dates of the study period so that it extends from August 2019 through January 2020. All calculations in the manuscript have been updated using this new 6-month study period, and the relevant figures have been modified accordingly.

The reviewer makes a fair point that the 2019 season may not be representative of Northern Australia on average. The TROPOMI data was released in 2018, so data for only two fire seasons (2018/2019 and 2019/2020) were available at the time that this study was initiated. We chose to use the 2019/2020 fire season because it provided the opportunity to quantify emission ratios and emission factors over both temperature forest fires (Area 3) and savanna fires (Areas 1 and 2). That would not have been possible with data from the 2018/2019 fire season because there was minimal biomass burning in Area 3 during that season.

We edited the sentence in L114,

"In Section 4, we report the fire intensity, and daily maximum and mean $NO_2$ and CO column densities observed during 6 months in 2019 and 2020 (i.e., 1 August 2019 to 31 January 2020) over fire hotspot regions."

And sentence in L128,

"The vegetation fires that happened in Australia from August 2019 through January 2020 were classified as savanna and temperate forest fires based on GFED4s."

We also added a sentence in L132,

"To be consistent for the three areas, we chose the same study period that covers all fires from August 2019 through January 2020."

Modified Figure 1 and Figures 3 to 5 in revision are:

[Figure]

**Figure 1** Total fire counts from August 2019 through January 2020 at $0.5^{\circ} \times 0.5^{\circ}$ resolution. Three $10^{\circ} \times 10^{\circ}$ (latitude $\times$ longitude) areas indicate the regions of interest in this study.

[Figure]

**Figure 3** Daily fire radiative power (FRP) from August 2019 through January 2020 for Area 1 (green), Area 2 (blue), and Area 3 (red). Several distinct periods are highlighted to show a significant increase in FRP, covering 1-24 October and 1 November - 3 December (Area 1), 4 – 13 September (Area 2), 7 - 18

November and 28 November - 29 December (Area 2), 5 - 17 November (Area 3), and 28 December - 6 January (Area 3).

[Figure]

**Figure 4** Monthly average $NO_2$ (a-f) and CO (g-l) column density from August 2019 through January 2020. Three $10^o \times 10^o$ (latitude $\times$ longitude) areas (red squares) indicate the regions of interest in this study.

[Figure]

**Figure 5** Time series of daily maximum $NO_2$ (a) and CO (b) total column densities from August 2019 through January 2020 as well as daily mean $NO_2$ (c) and CO (d) for three highlighted areas: Area 1 (green), Area 2 (blue), and Area 3 (red). Results for Areas 1 and 2 are displayed by the left Y axis and results for Area 3 are displayed by red colors on the right Y axis.

Sentences in Sec.4.1, L234 - 242 have been changed,

"The majority of fire-affected regions during these extreme fire events were located in Area 3 in southeast Australia (Fig. 1) where the largest cumulative fire counts exceeded 3,000. Fire frequencies were much lower in Areas 1 and 2 where the largest cumulative fire counts did not exceed 700. The fire-affected areas were dominantly located either in the far northern oceanic boundaries of Areas 1 and 2 or in the south-eastern oceanic boundary of Area 3 (Fig. 1). From the fire data product of MCD14DL, the daily FRP observations showed a few distinct periods of peak fire events (Fig. 3), including three weeks from October 1st to 24th and four weeks from November 1st to December 3rd in Area 1, and three weeks for Area 2 from November 28th to December 29th. For Area 3, there were two short FRP peaks in November and early January. The highest FRPs during these periods of peak fire events were $4.45\times10^4$, $4.44\times10^4$, $1.01\times10^6$ MJ s$^{-1}$ for Areas 1, 2, and 3, respectively. The most intensive fire events were observed in October and November 2019 for Area 1, in December 2019 for Area 2, and in January 2020 for Area 3 (Fig. 3)."

And, we also rewrote sentences in L248,

"The maxima of the daily column densities were observed as 1.26 mmol m$^{-2}$ for $NO_2$ on 28th November, and 2.3 mol m$^{-2}$ for CO on 4 January in Area 3."

2. L172: Concerning the appropriateness of an 850 hPa average height when aerosol layer height is unavailable. Please explain why it is appropriate to use 850 hPa as an average for all three regions. There were some very large pyrocumulus events in the 2019/2020 fire season. For example, does the average change appreciably if the pyrocumulus events are removed? Additionally, is 850 hPa used for Northern Australia, where pyrocumulus are rarer?

**Response:** Thank you for your comments. We agree that the use of one average value (i.e., 850 hPa) for all three areas may not be appropriate due to the differences in fire intensity over three areas. To address this issue, we have used plume height data from the GFAS (Global Fire Assimilation System, https://www.ecmwf.int/en/forecasts/dataset/global-fire-assimilation-system) as an additional data source for ALH. In our revision, the mean value of all available GFAS plume height data in each area (822 hPa for Area 1, 866 hPa for Area 2, and 833 hPa for Area 3) was used when both TROPOMI ALH and GFAS data were unavailable (Fig. R1).

[Figure]

**Figure R1** Plume height distribution over three areas using GFAS data and TROPOMI ALH dataset from August 2019 through January 2020 in our revision.

Accordingly, we rewrote Sec. 2.2's title as "TROPOMI CO, NO$_2$, and fire plume data", and the sentences in L147,

"The ALH data were used to define the main vertical wind layer which was required for the emission estimation procedure described in Section 3.2, and we added plume height data from the Global Fire Assimilation System (GFAS) as alternative values to use when ALH data were unavailable."

And in L172,

"For fire events without valid ALH data, the GFAS plume height data were used as a replacement. Otherwise, an average plume height over each area was used when both ALH and GFAS datasets were unavailable. The mean plume height was 822 hPa for Area 1, 866 hPa for Area 2, and 833 hPa for Area 3."

Thank you for pointing out pyrocumulus events. The pyrocumulonimbus (PyroCb) outbreak occurred during 29-31 December and 4 January 2020 in southeastern Australia during our study period (Kablick III et al., 2020; https://www.nrlmry.navy.mil/PyroCb_docs/htdocs/australia.html). Fig. R2 shows the daily CO and NO$_2$ distributions for 29-31 December and 4 January 2020. It also includes a map from Peterson et al. (2021) that shows the blow-up fires contributing to PyroCb activity. We found that most PyroCb events were removed in TROPOMI NO$_2$ and CO data during quality filtering process (by qa_value), which means only a few pixels over PyroCb events are included in the calculation. When we summed up the daily CO and NO$_2$ over fire pixels in Area 3 with/without PyroCb events for savanna and temperate forest vegetation (Tables R1 and R2), we found the values don't change significantly, which makes sense because the burning areas of PyroCb event in Fig. R2 (i) are small (the largest burning area is 1393 km$^2$ based on the PyroCb activity data provided by Dr. Peterson (personal comm.), around 7×7 pixels) compared to the whole of Area 3 (200×200 pixels). In addition, the average of available plume height doesn't change much when excluding PyroCb events days (838 hPa). We masked the areas that included PyroCb events when estimating the biomass burning CO and NO$_2$ emission because the flux method should be used under no PyroCb development condition (Griffin et al., 2021).

**Table R1** Comparison of CO and NO$_2$ emission with/without PyroCb events over fire regions in Area 3 for savanna vegetation. NO$_2$ used here is from the new TROPOMI NO$_2$ data in revision.

| Gas \ Date | CO (mol m$^{-2}$) | | NO$_2$ (mmol m$^{-2}$) | |
|---|---|---|---|---|
| | With PyroCb | Without PyroCb | With PyroCb | Without PyroCb |
| 2019-12-29 | 0.054 | 0.054 | 0.034 | 0.034 |
| 2019-12-30 | 0.069 | 0.069 | 0.035 | 0.033 |
| 2019-12-31 | 0.051 | 0.051 | NA | NA |
| 2020-01-04 | 0.101 | 0.960 | 0.057 | 0.053 |

**Table R2** Same as Table R1 but for temperate forest.

| Gas \ Date | CO (mol m$^{-2}$) | | NO$_2$ (mmol m$^{-2}$) | |
|---|---|---|---|---|
| | With PyroCb | Without PyroCb | With PyroCb | Without PyroCb |
| 2019-12-29 | 0.072 | 0.072 | 0.034 | 0.034 |
| 2019-12-30 | 0.094 | 0.091 | 0.048 | 0.047 |
| 2019-12-31 | 0.077 | 0.075 | NA | NA |
| 2020-01-04 | 0.245 | 0.219 | 0.084 | 0.084 |

[Figure]

**Figure R2** TROPOMI $NO_2$ and CO distribution during 29-31 December and 4 January 2020 (a-h), and the (i) map of the blow-up fires contributing to PyroCb activity from Peterson et al. (2021). The color-shadings are perimeters of blow-up fires. The red box denotes Area 3, the yellow box is the same area as in (i).

Then authors rewrote the sentences in Sec. 3.2, L195,

"When estimating CO and $NO_2$ emission from biomass burning, we excluded the TROPOMI dataset over the areas with pyrocumulus (PyroCb) events between 29-31 December 2019 and 4 January 2020 based on the PyroCb activity dataset of Peterson et al. (2021) because the flux method should be used under no PyroCb development condition (Griffin et al., 2021). Fire pixels were grouped based on distance as described in Section 2.3 and surrounding rectangles were defined."

3. Please clarify why it is appropriate to compare total column CO and tropospheric $NO_2$? For example, can you be confident you are capturing the same air masses.

**Response:** Wildfires are significant sources of tropospheric $NO_2$ (Crutzen, 1979) and some studies have shown that fire smoke plumes are often completely trapped within the atmospheric boundary layer (BL) (Trentmann et al., 2002), whereas major fires can provide sufficient energy to inject emissions above the BL (2 to 7km). During some uncommon events such as PyroCb events during 29-31 December and 4 January 2020, the fire smoke plume can even reach the lower stratosphere (>10 km) (Peterson et al., 2021). Some $NO_2$ emitted from fires may reach the lower stratosphere, but considering its short lifetime (only few hours), it may not be transported into the stratosphere. We do not use the total column $NO_2$ because we focus on the emission from fire and the fire plume is mainly in the tropospheric layer except under PyroCb events. Although the PyroCb events may influence emission analysis, most PyroCb events were excluded during our analysis (Tables R1 and R2). For CO, we chose to use the total column CO because CO emitted from fires can reach the low stratosphere (Hudson et al., 2004; Jost et al., 2004; Hooghiem et al., 2020) and has a long lifetime from days to months. Furthermore, other studies have shown that comparisons between the total column CO and tropospheric $NO_2$ during fire events are meaningful and useful (Lama et al., 2019; van der Velde et al., 2020).

4. Section 3.1
- Please describe or clarify how recirculating plumes are avoided in emission ratio calculations.

**Response:** Thank you for your insight. Recirculating plumes indeed cause uncertainties in our calculations. It is hard to exclude or isolate its influence in our study.

We added a sentence in Sec. 3.2, L227

"It should be noted that recirculating plumes have not been taken into account in our analysis, which may cause some degree of uncertainty in our emission ratio estimates."

- L183-184: Please clarify what specifically was used to determine upwind direction – a visual inspection of aerosol layer height, CO maps, ERA wind direction?

  **Response:** Thank you for your careful review. We used wind direction from ERA5 to determine upwind direction. Specifically, the ERA5 wind direction was interpolated at TROPOMI satellite overpass time (~ 13:30), and surface wind direction was used to determine upwind direction.

  We added the sentence in L183 for clarification,

  "The upwind direction was determined by interpolating the surface daily ERA5 wind data to the time and location of TROPOMI observations."

- CO and $NO_2$ also have strong anthropogenic sources – a comment about how this is accounted for would be valuable.

  **Response:** Thank you. we added a sentence in Sec.2.3, L188.

  "Although CO and $NO_2$ also have strong anthropogenic sources, we minimize the influence of anthropogenic sources by selecting pixels collocated with FRP pixels."

5. L191-192: Were there fewer fires in Northern Australia during 2018/2019 compared to 2019/2020?

**Response:** In Areas 1 and 2, there were larger fires during 2018/2019 than in 2019/2020 (Fig. R3). In the original manuscript, we chose 2018-2019 as background emission because the TROPOMI data was first released in 2018, so 2018-2019 was the only period available to use for background emission at the time we initiated our study. Because there were fewer fire events in 2019/2020, and TROPOMI data for 2019/2020 are now available, we have reanalyzed our results using background data from August 2020 through January 2021. The background values over three areas are relatively consistent (Table R3).

Based on our recalculations, we changed sentences in Sec.3.2 in L190 as,

"In our study, we used an integrated mass enhancement method that has been used in previous studies (e.g., Mebust et al., 2011; Adams et al., 2019; Griffin et al., 2021) to estimate downwind flux. Since the 2018-2019 fire events in Areas 1 and 2 were larger than those in 2020-2021, the period from August 2020 through January 2021 was used as the background data for both CO and $NO_2$ column densities, to represent emissions under less intense fire conditions. To improve background robustness for daily gas column density, we removed raw column density values that were above the 99th percentile on each day in each area, then refilled back by using the nearest neighboring data by interpolations. The mean value and standard deviations of background data indicated that the background selected did not have a strong systematic variation. The background values for CO ranged from $0.018 \pm 0.001$ to $0.032 \pm 0.002$ mol m$^{-2}$, and the background values for $NO_2$ ranged from $0.007 \pm 0.002$ to $0.011 \pm 0.005$ mmol m$^{-2}$."

[Figure]

**Figure R3** The daily FRP over three areas from August through January in 2018-2019, 2019-2020, and 2020-2021

**Table R3** The minima and maxima of background CO and $NO_2$ mean values among three areas, the value in the bracket is the standard deviation. $NO_2$ here represents new TROPOMI $NO_2$ data used in our revision.

| | CO range (mol/m$^2$) | $NO_2$ range (mmol/m$^2$) |
|---|---|---|
| August | 0.022(0.001) - 0.024(0.001) | 0.008(0.001) - 0.010(0.008) |
| September | 0.027(0.001) - 0.029(0.002) | 0.008(0.001) - 0.009(0.002) |
| October | 0.028(0.001) - 0.032(0.002) | 0.008(0.002) - 0.009(0.005) |
| November | 0.025(0.001) - 0.032(0.001) | 0.008(0.012) - 0.011(0.005) |
| December | 0.020(0.001) - 0.025(0.001) | 0.008(0.002) - 0.010(0.005) |
| January | 0.018(0.001) - 0.021(0.001) | 0.007(0.002) - 0.008(0.003) |

6. Section 3.2: I seem to have missed the description of the rotation of wind directions to align the pollution plume, as shown in Figure 2 c).

**Response:** Thank you for pointing out this question. We modified the sentences in L214 for clarification,

 "Figure 2 (a) - (c) shows an example of emission calculation for a fire event that occurred in Area 3 of south-eastern Australia (29.2 ºS, 151.5 ºE) on 6 November 2019, where the FRP fire pixels were grouped and TROPOMI data background column density values were removed. The location for the center of the fire was set at the averaged latitude and longitude of all fire pixels (the red star), then the mean wind direction was calculated. Lastly, the TROPOMI data plume direction (red arrow) was rotated to align with the wind direction."

And below is the revised Figure 2 in our revision,

[Figure]

**Figure 2** An example of emission analysis for a fire event, with MODIS fire pixels indicated (black points) and the center of the fire event indicated by a red star. (a) Map of TROPOMI NO$_2$ column density over Australia on 6 November 2019. The red box in southeast Australia marks the fire event location. (b) The original TROPOMI NO$_2$ column density with the wind direction is indicated by a white arrow. The red arrow indicates the plume direction. (c) The excess NO$_2$ after 1) removing background column density from original NO$_2$ and 2) rotating the entire pixels examined to align with the wind direction, thus a 20 km downwind distance area was selected and used to estimate the NO$_2$ emission.

7. Describe the "grand" or overall emissions ratio calculation in section 3.1.

**Response:** We added the sentence L188, "And the overall emission ratio for each area was calculated by averaging the daily emission ratios in the studied area."

8. Figure 9 – please add a description of the different color lines to the figure.

**Response:** Thank you. We updated Figure 9 and rewrote the caption in L370.

[Figure]

**Figure 9:** NO$_2$ line density decay curves of three example fire events (each color represents a fire event) in Area 2. The embedded histogram shows the frequency distribution of NO$_2$ lifetime estimated from all three areas.

9. Why was the Griffin et al. (2021) aerosol correction not applied to TROPOMI NO$_2$ retrievals here? It seems like this would improve the results for the NO$_2$ emission ratios.

**Response:** The main reason we didn't use the Griffin et al. (2021) aerosol correction on TROPOMI NO$_2$ retrievals is that there were relatively large amounts of missing ALH data after using the quality filtering. Instead, we recalculated emission ratio and emission factor by replacing the TROPOMI NO$_2$ data in the original manuscript with van Geffen et al. (2022)'s TROPOMI NO$_2$ data, which indicates that on average, the corrected NO$_2$ tropospheric vertical column densities were 10% to 40% larger than the raw data, especially over large, polluted regions.

We rewrote the sentences in L141,

"We chose an improved NO$_2$ dataset from van Geffen et al. (2022), which showed that on average, the corrected NO$_2$ tropospheric vertical column densities are 10% to 40% larger than the raw data, especially over large, polluted regions. Different algorithms that use different channels are used to estimate the NO$_2$ and CO from TROPOMI, and these algorithms internally provide quality assurance values (i.e., qa_value) to help filter raw data under unclear sky conditions and/or for other problematic retrievals."

And the sentences in L405 - 410,

"TROPOMI CO, ALH data are available from NASA Goddard Earth Sciences (GES) Data and Information Services Center (DISC, https://disc.gsfc.nasa.gov/datasets/). TROPOMI NO$_2$ is available at https://data-portal.s5p-pal.com/. The GFAS fire plume data is available at https://ads.atmosphere.copernicus.eu/cdsapp#!/dataset/cams-global-fire-emissions-gfas?tab=overview."

After we switched to the dataset of van Geffen et al. (2022), we updated the emission ratio and emission factor analysis. The related sentences were correspondingly adjusted based on updated results in original Figs. 6-8.

[Figure]

**Figure 6** The relationship between daily ΔCO (mol m$^{-2}$) and daily ΔNO$_2$ (mmol m$^{-2}$) in Savanna regions (a for Area 1, b for Area 2, and c for Area 3) and temperate forest regions (d for Area 3 only). The color bars are coded by daily FRP, data points with black edges are the days with high FRP (highlighted periods) in Figure 4. The blue markers represent the monthly average relationship between ΔCO and ΔNO$_2$ with day-to-day variabilities shown represented by the error bars. ER stands for the total emission ratio expressed by overall mean plus and minus one standard deviation.

[Figure]

**Figure 7** The relationship between daily ΔCO (mol m$^{-2}$) and daily ΔNO$_2$ (mmol m$^{-2}$) was derived from TROPOMI data for all areas. The slope of linear fit with an intercept at zero represents the combustion efficiency of different fire types.

[Figure]

**Figure 8** Scatter plots of TROPOMI-derived $NO_x$ and CO emissions (g s$^{-1}$) versus MODIS FRP in Savanna regions (a, d for Area 1, b, e for Area 2) and temperate forest regions (c, f for Area 3). The black line indicates the regression line estimated from ordinary least squares regression with the intercept fixed at zero. Slopes are shown with a 95% confidence interval. The color represents the plume height of the corresponding fire events. Emissions and FRP are on log scales.

Revised sentences in L279 now are,

"Different from the calculation of gas concentrations, the excess gas concentration (expressed as $\Delta X$) is derived by removing the impact of potentially varying amounts of background concentration, thus it represents the gas emissions related to fire activity only. The averaged ERs derived from savanna fires were 2.34, 2.60, and 2.03 for Areas 1, 2, and 3, respectively. The ER for temperate forests in Area 3 was, on average, 1.57 during the six months we studied (Fig. 6)."

And we also rewrote the sentences in L325

"The FRP explains 40% to 56% variance in $NO_x$ emissions with the lowest $R^2$ in temperate fires in Area 3 and the highest in savanna fires in Area 1. For CO emission, the FRP explained 35% to 51% variance with the highest $R^2$ in savanna fires and the lowest in temperate fires."

In addition, we rewrote L328,

"Comparing different fire types, the $NO_x$ emission coefficient in savanna fires in Area 1 is the largest (0.98 g MJ$^{-1}$), with 95% confidence intervals of 0.9 – 1.06 g MJ$^{-1}$. The CO emission coefficient in temperate forest fires in Area 3 is the largest (55.93 g MJ$^{-1}$), with 95% confidence intervals of 50.7 – 61.17 g MJ$^{-1}$."

And sentence L379,

"For $NO_x$, the satellite-derived EFs range from 1.48 to 2.39 g kg$^{-1}$ in savanna fires which are agreeable with previous studies (1.36 g kg$^{-1}$) in the Jin et al. (2021) using original TROPOMI $NO_2$ data without updating a priori profile but lower than the work in Andreae (2.5±1.3 g kg$^{-1}$) (2019) that presented the updated compilation of EFs over the past 20 years. For temperate forests, the satellite-derived EF$_{NOx}$ is 1.51 g kg$^{-1}$, which is also less than Andreae's EFs (3 g kg$^{-1}$) (2019). For CO, the satellite-derived EF$_{CO}$ in savanna fires ranges from 107.39 to 126.32 g kg$^{-1}$ and is larger than Andreae's EFs (69 ± 20 g kg$^{-1}$) (2019) but in the range of the field measurement (ranging from 15 to 147 g kg$^{-1}$) from SAFIRED campaign savanna fires in Australia (Desservettaz et al., 2017) Our satellite-derived EF$_{CO}$ in temperate forest fires is 136.41 g kg$^{-1}$ which is larger than Andreae's EFs (113 ± 50 g kg$^{-1}$) (2019) and Guérette's field measurement (ranging 101 to 118 g kg$^{-1}$) in Australia temperate forest fires (Guérette et al., 2018)."
* * *
**Technical Corrections:**
L109: seldom → sparse
L306: does "grand emission ration" mean "overall emission ratio"?
L341: filed → field

**Response:** Authors have corrected them.

**Reference** (* indicates that the reference was also added to the revised manuscript)

Adams, C., McLinden, C. A., Shephard, M. W., Dickson, N., Dammers, E., Chen, J., Makar, P., Cady-Pereira, K. E., Tam, N., Kharol, S. K., Lamsal, L. N., and Krotkov, N. A.: Satellite-derived emissions of carbon monoxide, ammonia, and nitrogen dioxide from the 2016 Horse River wildfire in the Fort McMurray area, Atmos. Chem. Phys., 19, 2577–2599, https://doi.org/10.5194/acp-19-2577-2019, 2019.

Andreae, M. O.: Emission of trace gases and aerosols from biomass burning – an updated assessment, Atmos. Chem. Phys., 19, 8523–8546, https://doi.org/10.5194/acp-19-8523-2019, 2019.

Crutzen, P. J.: The Role of NO and $NO_2$ in the Chemistry of the Troposphere and Stratosphere, Annu Rev Earth Planet Sci., 7, 443–472, https://doi.org/10.1146/annurev.ea.07.050179.002303, 1979.

Desservettaz, M., Paton-Walsh, C., Griffith, D. W. T., Kettlewell, G., Keywood, M. D., Vanderschoot, M. V., Ward, J., Mallet, M. D., Milic, A., Miljevic, B., Ristovski, Z. D., Howard, D., Edwards, G. C., and Atkinson, B.: Emission factors of trace gases and particles from tropical savanna fires in Australia, J. Geophys. Res. Atmos., 122, 6059–6074, https://doi.org/10.1002/2016JD025925, 2017.

Griffin, D., McLinden, C. A., Dammers, E., Adams, C., Stockwell, C. E., Warneke, C., Bourgeois, I., Peischl, J., Ryerson, T. B., Zarzana, K. J., Rowe, J. P., Volkamer, R., Knote, C., Kille, N., Koenig, T. K., Lee, C. F., Rollins, D., Rickly, P. S., Chen, J., Fehr, L., Bourassa, A., Degenstein, D., Hayden, K., Mihele, C., Wren, S. N., Liggio, J., Akingunola, A., and Makar, P.: Biomass burning nitrogen dioxide emissions derived from space with TROPOMI: methodology and validation, Atmos. Meas. Tech., 14, 7929–7957, https://doi.org/10.5194/amt-14-7929-2021, 2021.

Guérette, E.-A., Paton-Walsh, C., Desservettaz, M., Smith, T. E. L., Volkova, L., Weston, C. J., and Meyer, C. P.: Emissions of trace gases from Australian temperate forest fires: emission factors and dependence on modified combustion efficiency, Atmos. Chem. Phys., 18, 3717–3735, https://doi.org/10.5194/acp-18-3717-2018, 2018.

Hooghiem, J. J. D., Popa, M. E., Röckmann, T., Grooß, J.-U., Tritscher, I., Müller, R., Kivi, R., and Chen, H.: Wildfire smoke in the lower stratosphere identified by in situ CO observations, Atmos. Chem. Phys., 20, 13985–14003, https://doi.org/10.5194/acp-20-13985-2020, 2020.

Hudson, P. K., Murphy, D. M., Cziczo, D. J., Thomson, D. S., de Gouw, J. A., Warneke, C., Holloway, J., Jost, H.-J., and Hübler, G.: Biomass-burning particle measurements: Characteristic composition and chemical processing, J. Geophys. Res. Atmos., 109, https://doi.org/10.1029/2003JD004398, 2004.

Jost, H.-J., Drdla, K., Stohl, A., Pfister, L., Loewenstein, M., Lopez, J. P., Hudson, P. K., Murphy, D. M., Cziczo, D. J., Fromm, M., Bui, T. P., Dean-Day, J., Gerbig, C., Mahoney, M. J., Richard, E. C., Spichtinger, N., Pittman, J. V., Weinstock, E. M., Wilson, J. C., and Xueref, I.: In-situ observations of mid-latitude forest fire plumes deep in the stratosphere, Geophys. Res. Lett., 31, https://doi.org/10.1029/2003GL019253, 2004.

Jin, X., Zhu, Q., and Cohen, R.: Direct estimates of biomass burning $NO_x$ emissions and lifetime using daily observations from TROPOMI, Atmos. Chem. Phys., https://doi.org/10.5194/acp-2021-381, 2021.

Kablick III, G. P., Allen, D. R., Fromm, M. D., and Nedoluha, G. E.: Australian PyroCb Smoke Generates Synoptic-Scale Stratospheric Anticyclones, Geophys. Res. Lett., 47, e2020GL088101, https://doi.org/10.1029/2020GL088101, 2020.

Lama, S., Houweling, S., Boersma, K. F., Aben, I., van der Gon, H. A. C. D., Krol, M. C., Dolman, A. J., Borsdorff, T., and Lorente, A.: Quantifying burning efficiency in Megacities using $NO_2$/CO ratio from the Tropospheric Monitoring Instrument (TROPOMI), Atmos. Chem. Phys., https://doi.org/10.5194/acp-2019-1112, 2019.

Mebust, A. K., Russell, A. R., Hudman, R. C., Valin, L. C., and Cohen, R. C.: Characterization of wildfire NOx emissions using MODIS fire radiative power and OMI tropospheric $NO_2$ columns, Atmos. Chem. Phys., 11, 5839–5851, https://doi.org/10.5194/acp-11-5839-2011, 2011.

*Peterson, D. A., Fromm, M. D., McRae, R. H. D., Campbell, J. R., Hyer, E. J., Taha, G., Camacho, C. P., Kablick, G. P., Schmidt, C. C., and DeLand, M. T.: Australia's Black Summer pyrocumulonimbus super outbreak reveals potential for increasingly extreme stratospheric smoke events, NPJ Clim Atmos Sci, 4, 1–16, https://doi.org/10.1038/s41612-021-00192-9, 2021.

Russell-Smith, J. and Yates, C. P.: Australian Savanna Fire Regimes: Context, Scales, Patchiness, Fire Ecol, 3, 48–63, https://doi.org/10.4996/fireecology.0301048, 2007.

Trentmann, J., Andreae, M. O., Graf, H.-F., Hobbs, P. V., Ottmar, R. D., and Trautmann, T.: Simulation of a biomass-burning plume: Comparison of model results with observations, J. Geophys. Res. Atmos., 107, AAC 5-1-AAC 5-15, https://doi.org/10.1029/2001JD000410, 2002.

van der Velde, I. R., van der Werf, G. R., Houweling, S., Eskes, H. J., Veefkind, J. P., Borsdorff, T., and Aben, I.: Biomass burning combustion efficiency observed from space using measurements of CO and $NO_2$ by TROPOMI, Atmos. Chem. Phys., https://doi.org/10.5194/acp-2020-272, 2020.

*van Geffen, J., Eskes, H., Compernolle, S., Pinardi, G., Verhoelst, T., Lambert, J.-C., Sneep, M., ter Linden, M., Ludewig, A., Boersma, K. F., and Veefkind, J. P.: Sentinel-5P TROPOMI $NO_2$ retrieval: impact of version v2.2 improvements and comparisons with OMI and ground-based data, Atmos. Meas. Tech., 15, 2037–2060, https://doi.org/10.5194/amt-15-2037-2022, 2022.

---

## Author Comment (AC2)

**Point-by-point Responses to Reviewer #2 for # MS No.: acp-2022-447**

The authors have used regular fonts for the Referee's comments (which might be divided into two or multiple comments), blue fonts for our responses, and red fonts with quotation marks to show the revised text.

**Reviewer 2 # Comments**

Overall, this is an excellent research article. In addition to the suggested revisions of the other reviewer, I would like to see improvement in the abstract as well as the summary and conclusions, which are a bit thin on the important implications of your research. For instance, you make the following statement in the summary and conclusions: "Our study on both savanna and temperate forest fire emissions demonstrates the capability and limitations of TROPOMI data for the study of the regional variability of combustion characteristics and their impacts on regional atmospheric composition and air quality." You make a similar comment in the abstract. This statement may be accurate, but I would like you to elaborate on this statement, including on how your technique may be applied to other world regions. As another example, you say: "These differences could be traced back to different measurement techniques used, their spatial resolutions, nonlinear sensitivities to gas densities in the boundary layer, and larger NO2 natural variability due to its short lifetime, all of which suggest that further validation of satellite products and investigations of more cases are required." Could you suggest additional validation that would be most helpful to this end? How many cases are required? My recommendation is to revisit the abstract and summary and conclusions with an eye for elaborating on the broader implications of your research.

**Response:** Thank you for your comments. We rewrote the abstract,

[revised manuscript text omitted]